# Quantum and Classical Cosmology in the Brans–Dicke Theory

Carla R. Almeida [1,*] , Olesya Galkina [1] and Julio César Fabris [1,2]

1   Núcleo Cosmo-ufes & Departamento de Física, Universidade Federal do Espírito Santo,
    Vitória 29075-910, ES, Brazil; olesya.galkina@cosmo-ufes.org (O.G.); julio.fabris@cosmo-ufes.org (J.C.F.)
2   National Research Nuclear University MEPhI, Kashirskoe sh. 31, 115409 Moscow, Russia
*   Correspondence: cralmeida00@gmail.com

**Abstract:** In this paper, we discuss classical and quantum aspects of cosmological models in the Brans–Dicke theory. First, we review cosmological bounce solutions in the Brans–Dicke theory that obeys energy conditions (without ghost) for a universe filled with radiative fluid. Then, we quantize this classical model in a canonical way, establishing the corresponding Wheeler–DeWitt equation in the minisuperspace, and analyze the quantum solutions. When the energy conditions are violated, corresponding to the case $\omega < -\frac{3}{2}$, the energy is bounded from below and singularity-free solutions are found. However, in the case $\omega > -\frac{3}{2}$, we cannot compute the evolution of the scale factor by evaluating the expectation values because the wave function is not finite (energy spectrum is not bounded from below). However, we can analyze this case using Bohmian mechanics and the de Broglie–Bohm interpretation of quantum mechanics. Using this approach, the classical and quantum results can be compared for any value of $\omega$.

**Keywords:** Brans–Dicke theory; bounce models; de Broglie–Bohm interpretation





## 1. Introduction

General Relativity (GR) theory is a highly successful theory that describes gravitational interactions of the universe. It has successfully survived observational tests in the solar system, and its predictions about the emission of gravitational waves by binary systems were confirmed. However, there are strong indications that the theory is incomplete. For example, it cannot account for the observed gravitational anomalies, mainly in the cosmological context, and thus it is necessary to introduce the hypothesis of dark matter and dark energy to explain cosmological observational data within GR's framework. Another reason is the initial singularity at the beginning of the universe and at the final stage of some class of stars, which are generally predicted by the GR theory. These drawbacks led to many possible extensions of the GR that can address those problems. The prototype of alternative theory of gravity is Brans–Dicke theory (BD). Historically, it is one of the most important modifications to the standard GR theory, which was introduced by Brans and Dicke [1] as a possible implementation of Mach's principle in a relativistic theory (eventually, it did not work out as we will discuss later).

It is expected that a quantum formulation of GR might solve some of its problems, especially those related to the existence of singularities. However, there are many obstacles to this quantization [2,3]. Several attempts have been made to overcome the difficulties that appear when combining the principles of the GR theory and Quantum Mechanics (QM), be it via canonical methods or other procedures, like loop quantization or string theory. A simplified approach, like the quantization of the Einstein–Hilbert action in the minisuperspace in the presence of matter fields, shows that it is possible to obtain cosmological models without singularities. The construction of a quantum cosmological model encounters many problems, even when the minisuperspace restriction is used. The first one is the absence of an explicit time coordinate due to the invariance by time reparametrizations in the classical theory [4,5]. There are different ways to solve this problem. One of them

is to allow the matter fields to play the role of time, which can be achieved, for example, through Schutz's description of a fluid [6]: its corresponding canonical formulation results in a Schrödinger-type equation since the conjugate momentum associated with the matter variables appears linearly in the Hamiltonian. We will employ this approach in the analysis to be exposed in the present text.

Another point of discussion is a choice of the suitable formalism to interpret the quantum theory and thus obtain specific predictions. This is a very sensible question. The usual Copenhagen interpretation is based on a probabilistic formalism, using concepts such as decoherence and a measurement mechanism through the spectral theorem. It is not ideal for a system consisting of a unique realization as it is the universe. Nevertheless, many adaptations of the Copenhagen interpretation are possible, like the Many World [7] or the Consistent Histories [8]. One alternative is the de Broglie–Bohm (dBB) interpretation of quantum mechanics [9,10], one we will address in this work. The dBB approach keeps the concept of trajectories of a given system, and a probabilistic analysis is not fundamental in this scheme [11].

In this work, we will investigate the BD theory. The reason to analyze this theory, which has been thoroughly studied in many contexts, is that it reserves some interesting and even unexpected features which deserve to be discussed in more detail. Classically, cosmological models constructed from the BD theory may lead to non-singular scenarios if the energy conditions are violated, as it happens in the GR theory. Such violations of the energy conditions occur when the Brans–Dicke parameter varies in the domain of $\omega < -3/2$. Surprisingly, it is also possible to obtain a non-singular model in the BD theory even if the energy conditions are satisfied. We will discuss this possibility using radiation as the matter content. The situation becomes more complex when we turn to quantum models: a consistent quantum model, from the point of view of the Copenhagen interpretation, is possible when $\omega < -3/2$, since only in this case is the energy bounded from below; if the energy conditions are satisfied, the spectrum of energy is not bounded from below. However, in both cases, the analysis becomes possible even if we use the dBB interpretation of quantum mechanics. Moreover, the use of the dBB formalism allows a comparison between the classical and quantum models in the BD theory, at least to some simple configurations. This possibility will be explored in the present work, revealing many peculiarities at the classical and quantum levels.

The paper is organized as follows: in Section 2, we review the classical BD theory. In Section 3, we present a non-singular model with radiative fluid in BD theory that obeys the energy conditions and does not contain ghosts. Section 4 presents the quantization of the model with a Lagrangian with a non-minimally coupled scalar field and matter fluid, establishing the corresponding Wheeler–DeWitt equation in the minisuperspace. Finally, we analyze the quantum solution via dBB interpretation in Section 5. Our conclusions are revealed in Section 6.

## 2. A Short Review of the Brans–Dicke Theory

The Brans–Dicke theory was proposed as a modification of the relativistic theory of gravity to include two new features: the possibility of a dynamical coupling to gravity and Mach's principle. Dirac had suggested earlier that the gravitational coupling in Einstein's theory might not be a constant, implying it could vary with time, at least in the cosmological context [12]. This idea was further developed by Jordan [13], but the rigorous implementation was made by Brans and Dicke in their seminal 1961 article [1]. They replaced the gravitational coupling by the inverse of a scalar field $\phi$, such that

$$G \propto \frac{1}{\phi}. \tag{1}$$

A simple but elegant solution to implement this idea in a relativistic context is to consider in the action a non-minimal coupling between the new scalar field $\phi$ and the geometry represented by the Ricci scalar. The introduction of a kinetic term coupled by an arbi-

trary parameter $\omega$ complemented the theory, preserving the minimal coupling between gravity and matter and assuring the invariance by the full diffeomorphism group and the consequent conservation of the energy–momentum tensor.

The presence of a long-range scalar field was connected initially with the intention to implement the Mach's principle. The most common of many formulations of Mach's principle states that the inertial property of a given body results from its interaction with all matter present in the universe. It is not easy to implement such an appealing idea. General Relativity was considered to be a Machian theory since it relates the geometry of space-time to the matter distribution, but it has non-Machian features, such as the locality of the relation between space and matter and initial conditions for the matter fields. Perhaps, these issues could be circumvented by introducing a long-range scalar field as it occurs in the BD theory. In a Machian relativistic theory, the only solution in the absence of matter should be the Minkowski one. This is not the case for GR or for BD theory. In other words, both theories failed as a proposal to implement the Mach principle. However, introducing the scalar field $\phi$ non-minimally coupled to gravity led to many phenomenological applications and intriguing results: it opens new possibilities in comparison with the GR original framework.

The BD action reads as

$$S = \int d^4x \sqrt{-g} \left\{ \phi R - \frac{\omega}{\phi} \phi_{;\mu} \phi^{;\mu} \right\} + \int d^4x \sqrt{-g} \mathcal{L}_m(g_{\mu\nu}, \Psi), \tag{2}$$

where $\omega$ is a coupling constant and $\mathcal{L}_m$ is the matter term [1], with $\Psi$ indicating generically the matter fields.

It is commonly understood that, in the $\omega \to \infty$ limit, BD theory coincides with GR [14,15]. Although it is true in most situations, this statement is not valid in general. When $\omega \gg 1$, the field equations show that $\Box\phi = \mathcal{O}\left(\frac{1}{\omega}\right)$, so we have

$$\phi = \frac{1}{G_N} + \mathcal{O}\left(\frac{1}{\omega}\right), \tag{3}$$

$$G_{\mu\nu} = 8\pi G_N T_{\mu\nu} + \mathcal{O}\left(\frac{1}{\omega}\right), \tag{4}$$

where $G_N$ is Newton's gravitational constant and $G_{\mu\nu}$ is the Einstein tensor. However, there are some examples [16–18] where exact solutions cannot be continuously deformed into the corresponding GR solutions by taking the $\omega \to \infty$ limit. In this case, the solutions decay as

$$\phi = \frac{1}{G_N} + \mathcal{O}\left(\frac{1}{\sqrt{\omega}}\right). \tag{5}$$

Moreover, there is a particle solution that admits the appropriate asymptotic behavior given by Equation (3) but no GR limit [19].

From the action (2), the whole theory is derived. The field equations are obtained through the variation of the action (2) with respect to the metric and scalar field,

$$R_{\mu\nu} - \frac{1}{2} g_{\mu\nu} R = 8\pi \frac{T_{\mu\nu}}{\phi} + \frac{\omega}{\phi^2} \left( \phi_{;\mu} \phi_{;\nu} - \frac{1}{2} g_{\mu\nu} \phi_{;\alpha} \phi^{;\alpha} \right) + \frac{1}{\phi} (\phi_{;\mu;\nu} - g_{\mu\nu} \Box\phi), \tag{6}$$

$$\Box\phi = 8\pi \frac{T}{3 + 2\omega}. \tag{7}$$

The coupling with the curvature in the action (2) can be avoided if we perform a conformal transformation on the metric $g_{\mu\nu}$ such that

$$g_{\mu\nu} = \phi^{-1} \tilde{g}_{\mu\nu}. \tag{8}$$

With this, we change our frame of reference. With the reference frame $\tilde{g}_{\mu\nu}$, one could interpret the scalar field as a matter field, and thus recover general relativity. We, however, want to remain closer to Brans and Dicke's original idea of the gravitation not being purely geometrical, and thus the scalar field does not account for a matter content. The original frame is known as Jordan's frame, and, after the conformal transformation (8), it is called Einstein's frame. In this latter, the action becomes

$$S = \int d^4x \sqrt{-\tilde{g}} \left\{ \tilde{R} - \epsilon \zeta_{;\mu} \zeta^{;\mu} \right\} + \int d^4x \sqrt{-g} \mathcal{L}_m \left( e^{-\kappa\zeta} \tilde{g}_{\mu\nu}, \Psi \right), \tag{9}$$

where we have defined

$$\phi = e^{\kappa\zeta}, \tag{10}$$

$$\kappa = \frac{1}{\sqrt{\left| \omega + \frac{3}{2} \right|}}, \tag{11}$$

$$\epsilon = \text{sign}\left( \omega + \frac{3}{2} \right). \tag{12}$$

The resulting field equations are (ignoring the tildes in the redefined geometric terms),

$$R_{\mu\nu} - \frac{1}{2} g_{\mu\nu} R = 8\pi e^{-2\kappa\zeta} T_{\mu\nu} + \epsilon \left( \zeta_{;\mu} \zeta_{;\nu} - \frac{1}{2} g_{\mu\nu} \zeta_{;\rho} \zeta^{;\rho} \right), \tag{13}$$

$$\Box\zeta = \epsilon 4\pi T e^{-2\kappa\zeta}. \tag{14}$$

Please note that this set of equations implies a non-usual expression for the conservation laws, which is due to the non-minimal coupling between the scalar field and matter.

Thus, in the Einstein frame, the case $\epsilon = 1$ ($\omega > -\frac{3}{2}$) corresponds to an ordinary scalar field with positive energy density, while, for $\epsilon = -1$ ($\omega < -\frac{3}{2}$), the kinetic term of the scalar field changes sign, and it becomes a phantom field with negative energy density. In the special case $\omega = -3/2$ with a redefinition of the matter fields, the GR theory is recovered. When the matter field is given by radiation, the non-minimal coupling between the scalar field and the matter component is also broken since $T = 0$ for a radiative fluid. We will use this fact later.

Currently, the limits on the parameter $\omega$ are very stringent [20]. Cosmological constraints lead to values for $\omega$ of the order of some hundreds. Binary pulsars push these bounds to dozen of thousands. Hence, the BD theory essentially becomes GR in most of the cases, as mentioned before. Despite this, BD theory continues to be relevant for several reasons. For example, for a particular value of $\omega = -1$, the action (9) coincides with the effective string action for the dilatonic sector. In this case, the action (9) acquires some duality properties that have been explored in the Pre-Big Bang scenarios, in which there is a contracting phase in the evolution of the universe before the actual expanding phase. In fact, for $\omega = -1$, the action (2) in the cosmological context is invariant by the transformation,

$$a \to \frac{1}{a}, \quad \phi \to \phi - 2\ln a, \tag{15}$$

where $a$ is the scale factor. Therefore, an expanding universe can be mapped into a contracting universe. For an excellent recent review of Pre-Big Bang scenarios, we refer to [21]. Even if there is a clear discrepancy with the observational constraint on the value of $\omega$, it is possible to take into account a possible dependence of $\omega$ on energy scales that may reconcile observations with theoretical considerations. This may be achieved, for example, by supposing that $\omega$ is a function of the scalar field $\phi$, $\omega = \omega(\phi)$.

Perhaps, one of the most delicate aspects concerning the pre-big bang scenario deals with the evolution of the perturbations in the primordial universe, which becomes highly discrepant with GR and, at the same time, strictly constrained by observations. In general, predictions for the evolution of perturbations depend on the transition from the contracting phase to an expanding phase. A possible solution to this problem may be found by studying perturbations behavior, taking into account the string configuration underlining the pre-big bang model since the transition may occur at the large curvature regime. Such analysis is not an easy task, technically and conceptually. Similar proposals, like the ekpyrotic scenario based on brane-world structures coming from string theories, have been developed in the literature [22].

In what concerns the primordial universe, we may also mention the extended inflationary model using the original proposal based on a cosmological constant [23] but implemented in the context of the BD theory. It is possible to obtain the transition to the radiative phase, but only if the value of the parameter $\omega$ is relatively small, in contradiction with the constraints mentioned above. In such a case, it is necessary to find a mechanism to change the value of $\omega$ during the universe's evolution since the energy scale may depend on this parameter, as already evoked above.

Multidimensional theories, when compactified to four dimensions, leads to action similar to the BD, with

$$\omega = -\frac{d-1}{d}. \tag{16}$$

In this case, gauge fields can appear, depending on the original configuration, with non-trivial coupling [24,25].

The proposal of modified gravity theories spiked a revival in the interest in the BD theory. Modified gravity theories were conceived mainly to address the problem of the acceleration of the universe without introducing a new exotic component in the universe called the dark energy. There are many types of the modified gravity theories. For example, in the Horndesky theories, the most general Lagrangian includes the scalar field in a non-trivial way and leads to second-order equations of motion. In fact, the BD theory can be considered the first of the modified theories, and it was formulated long before the Horndesky classification. Another class of modified gravity theories is the $f(R)$ theories. They are based on a nonlinear generalization of the Einstein–Hilbert action. Interestingly, the $f(R)$ theories can be recast as the BD theory, with $\omega = 0$ and a potential term that depends on the form of the $f(R)$ function. In general, all modified gravity theories must introduce a screening mechanism to reconcile the large-scale and small-scale constraints. According to these screening mechanisms, the supplementary degree of freedom associated with the scalar field does not propagate in a dense region (compared to the averaged cosmological density). The BD theory, when reformulated in the Einstein frame, gives the prototype of the chameleon mechanism, one of the most important screening mechanisms: the conformal transformation used to reformulate the BD theory in the Einstein frame introduces a non-minimal coupling between matter and scalar field, making the mass of the scalar field depend on the density of the medium, as it can be verified introducing a potential term in the set of Equations (13) and (14).

Intriguing results on the quantization of the BD theory adds to the relevance of the theory. Analysis on quantum cosmological scenarios using the BD theory was made in Refs. [26,27]. In Ref. [26], the authors considered the Einstein frame and used the WKB formalism to recover the notion of time. The result reveals a curious behavior: the quantum effects become relevant in a late phase of the expansion of the universe, contrary to expectation. Due to this property, the initial singularity existing in the FLRW models can not be avoided. In [27], the study was extended to the Jordan frame through the Bohm–de Broglie formalism to compute the quantum evolution of the universe. In this case, the opposite scenario was found: the quantum effects become important mainly in the early universe, and the initial singularity can be avoided.

We must also mention the study of the resulting quantum Schrödinger-like equation from the point of view of its self-adjointness properties [28], which are essential to employ the spectral theorem. The results indicated that, for the case $\omega < -3/2$, which corresponds to a phantom scalar field, it is possible to recover the self-adjointness of the quantum operator. We will discuss this in more detail in the forthcoming sections.

### 3. Bouncing Solutions and the Energy Conditions

The most common solution to avoiding a singularity in cosmological models is the introduction of exotic types of matter fields, for example, a scalar field with negative energy density. On the other hand, to obtain a bouncing solution in classical General Relativity, violation of the energy conditions is required. In this section, after briefly reviewing the bouncing scenarios, we present a non-singular model with radiative fluid in BD theory that obeys the energy conditions and does not contain ghosts [29]. To do so, we first briefly analyze the solutions determined by Gurevich et al. [30] for the cosmological isotropic and homogeneous flat universe with a perfect fluid with an equation of state $p = n\rho$, where the parameter $n$ is given by $0 \leq n \leq 1$. Then, we focus on the analysis of these solutions in the case of radiative fluid.

The study of bouncing models is motivated by the search for cosmological solutions without singularities. Bouncing models have been widely studied to solve the initial-singularity problem and as an addition or alternative to inflation to describe the primordial universe because they can explain, in their own way, the horizon and flatness problems and justify the power spectrum of primordial cosmological perturbations inferred by observations [31–33].

In such models, an initial singularity is replaced with a bounce—a smooth transition from contraction to expansion. To obtain a bounce, one needs to change the value of the Hubble parameter $H \equiv \frac{\dot{a}(t)}{a(t)}$, which appears to be negative during the contracting phase, to a positive value for the following expanding phase. One of the options to switch the sign of the Hubble function lies within GR. It usually requires the violation of the null energy condition (NEC) [34]

$$\rho + p \geq 0, \tag{17}$$

where $\rho$ is energy density, and $p$ is pressure as usual. In most cases, the energy conditions reflect the nature of matter fields, "ordinary" (attractive effects) or "exotic" (repulsive effects). However, as we will see later, the bounce can be achieved through some non-standard coupling between the fields existing in a given theory.

In the early stages of the expansion of the universe, the curvature is not essential since the matter components are more relevant for the dynamic than the curvature term due to their dependence on the scale factor, and we can restrict our further analysis to the quasi-Euclidean variant of the isotropic model. In this case, for a flat FLRW metric,

$$ds^2 = N^2 dt^2 - a^2(t)\left(dx^2 + dy^2 + dz^2\right), \tag{18}$$

with $N^2 = 1$, the classical field Equations (6) and (7) reduce to

$$3\left(\frac{\dot{a}}{a}\right)^2 = 8\pi\frac{\rho}{\phi} + \frac{\omega}{2}\left(\frac{\dot{\phi}}{\phi}\right)^2 - 3\frac{\dot{a}}{a}\frac{\dot{\phi}}{\phi}, \tag{19}$$

$$2\frac{\ddot{a}}{a} + \left(\frac{\dot{a}}{a}\right)^2 = -8\pi\frac{p}{\phi} - \frac{\omega}{2}\left(\frac{\dot{\phi}}{\phi}\right)^2 - \frac{\ddot{\phi}}{\phi} - 2\frac{\dot{a}}{a}\frac{\dot{\phi}}{\phi}, \tag{20}$$

$$\ddot{\phi} + 3\frac{\dot{a}}{a}\dot{\phi} = \frac{8\pi}{3 + 2\omega}(\rho - 3p), \tag{21}$$

$$\dot{\rho} + 3\frac{\dot{a}}{a}(\rho + p) = 0. \tag{22}$$

The general solutions in [30] are obtained for $\omega < -\frac{3}{2}$ and $\omega > -\frac{3}{2}$. In the first case, there is violation of the energy conditions for the scalar field in Einstein's frame. In the latter case, the energy conditions for the scalar field are satisfied as we will discuss in this section. The solutions for the scale factor and scalar field for $\omega < -\frac{3}{2}$ are

$$a = a_0 \left[ (\theta + \theta_-)^2 + \theta_+^2 \right]^{\frac{\sigma}{2A}} e^{\pm \frac{\sqrt{\frac{2}{3}|\omega| - 1}}{A} \arctan \frac{\theta + \theta_-}{\theta_+}}, \tag{23}$$

$$\phi = \phi_0 \left[ (\theta + \theta_-)^2 + \theta_+^2 \right]^{(1-3n)/2A} e^{\mp 3(1-n) \frac{\sqrt{\frac{2}{3}|\omega| - 1}}{A} \arctan \frac{\theta + \theta_-}{\theta_+}}, \tag{24}$$

where $\sigma = 1 + \omega(1 - n)$, $2A = (1 - 3n) + 3\sigma(1 - n)$, $a_0$ is an arbitrary constant, and $\theta_- > \theta_+$ are integration constants. The time coordinate $\theta$ is connected with the cosmic time $t$ by definition,

$$dt = a^{3n} d\theta. \tag{25}$$

When $\theta \to \infty$, the scale factor $a$ does not vanish. The infinite contraction has a minimum $a_{min}$, and it is followed by the expansion. Thus, this model admits a cosmological bounce. For $\omega > -\frac{3}{2}$, the general solutions are

$$a(\theta) = a_0 (\theta - \theta_+)^{\omega/3(\sigma \mp \zeta)} (\theta - \theta_-)^{\omega/3(\sigma \pm \zeta)}, \tag{26}$$

$$\phi(\theta) = \phi_0 (\theta - \theta_+)^{(1 \mp \zeta)/(\sigma \mp \zeta)} (\theta - \theta_-)^{(1 \pm \zeta)/(\sigma \pm \zeta)}, \tag{27}$$

where $\zeta = \sqrt{1 + \frac{2}{3}\omega}$. In this model, a regular bounce can be obtained for $\frac{1}{4} < n < 1$ and $-\frac{3}{2} < \omega \leq -\frac{4}{3}$. The case $n = 1$ is unusual and does not admit bounce solution [19].

Henceforth, we shall focus on the universe filled with radiative fluid, represented by the equation of state $p = \frac{1}{3}\rho$. In this case, the solutions for $\omega > -\frac{3}{2}$ are given by the following expressions:

$$a(\eta) = a_0 (\eta - \eta_+)^{\frac{1}{2} \pm \frac{1}{2\sqrt{1 + \frac{2}{3}\omega}}} (\eta - \eta_-)^{\frac{1}{2} \mp \frac{1}{2\sqrt{1 + \frac{2}{3}\omega}}}, \tag{28}$$

$$\phi(\eta) = \phi_0 (\eta - \eta_+)^{\mp \frac{1}{\sqrt{1 + \frac{2}{3}\omega}}} (\eta - \eta_-)^{\pm \frac{1}{\sqrt{1 + \frac{2}{3}\omega}}}, \tag{29}$$

where $\eta$ is the conformal time and $\eta_\pm$ are constants such that $\eta_+ > \eta_-$. In Figure 1, we plot the scale factor and scalar field for the lower sign. The solutions for $\omega < -\frac{3}{2}$ are

$$a(\eta) = a_0 [(\eta + \eta_-)^2 + \eta_+^2]^{\frac{1}{2}} e^{\pm \frac{1}{\sqrt{\frac{2}{3}|\omega| - 1}} \arctan \frac{\eta + \eta_-}{\eta_+}}, \tag{30}$$

$$\phi(\eta) = \phi_0 e^{\mp \frac{2}{\sqrt{\frac{2}{3}|\omega| - 1}} \arctan \frac{\eta + \eta_-}{\eta_+}}. \tag{31}$$

In the case of the lower sign in Equations (28) and (29), we obtain bounce solutions for $-\frac{3}{2} < \omega < 0$. Nevertheless, there is a curvature singularity at $\eta = \eta_+$ for $-\frac{4}{3} < \omega < 0$, even if the scale factor diverges at this point. However, for $-\frac{3}{2} < \omega \leq -\frac{4}{3}$, bounce solutions are always regular without curvature singularity. In this case, there are two possible scenarios of the evolution of the universe due to the time reversal invariance. In the first one, the universe begins at $\eta = \eta_+$, with $a \to \infty$ and an infinite value for the gravitational coupling ($\phi = 0$). It evolves to the other asymptotic limit with $a \to \infty$, although with $\phi$ constant and finite. The second option is the reversal behavior of the first one for $-\infty < \eta < -\eta_+$. In both cases, the cosmic times varies as $-\infty < t < \infty$.

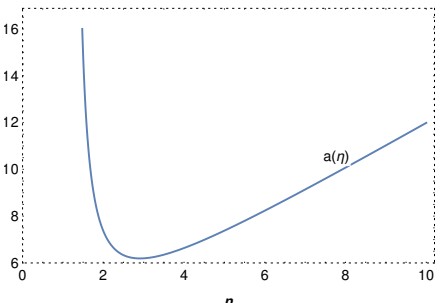
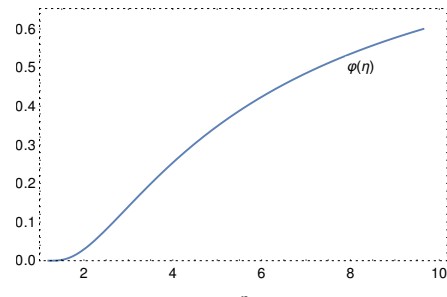

**Figure 1.** Behavior of the scale factor and scalar field in the case of the radiative fluid (Equations (28) and (29), lower sign) for $\omega = -1.43$.

The dual solution in the Einstein frame for $-\frac{3}{2} < \omega \leq -\frac{4}{3}$ is given by

$$b(\eta) = b_0(\eta - \eta_+)^{1/2}(\eta - \eta_-)^{1/2} \tag{32}$$

with $b = \phi^{1/2}a$ and contains an initial singularity. This can be considered as a specific case of "conformal continuation" in the scalar-tensor gravity proposed in [35].

Let us show that the energy conditions for the scalar field are satisfied for $\omega > -\frac{3}{2}$. In general, in order to have a bounce solution, violation of the energy conditions is required. Using the Friedmann and Raychaudhuri equations, we represent the strong and null energy conditions in GR as

$$\frac{\ddot{a}}{a} = -\frac{4\pi G}{3}(\rho + 3p) > 0 \tag{33}$$

$$-2\frac{\ddot{a}}{a} + 2\left(\frac{\dot{a}}{a}\right)^2 = 8\pi G(\rho + p) > 0. \tag{34}$$

We reformulate the BD theory in the Einstein frame so that it would be possible to use the energy condition in this form. One can see that both energy conditions are satisfied as long as $\omega > -\frac{3}{2}$. This is in agreement with the fact that, in the Einstein frame, the cosmological models are singular unless $\omega < -\frac{3}{2}$. However, in the original Jordan frame, there are non-singular models for $-\frac{3}{2} < \omega < -\frac{4}{3}$. However, in this range, the scalar field and the matter component obey the energy condition. The effects that lead to the absence of the singularity come from the non-minimal coupling. In Figure 2, we present the effective energy condition, defined in the left-hand side of Equations (33) and (34), considering the effects of the non-minimal coupling. If we analyze only the left-hand side in Equations (33) and (34), the effects of the interaction due to the non-minimal coupling are included, and the energy conditions can be violated even if the matter terms do not violate them. For more details, see Ref. [29].

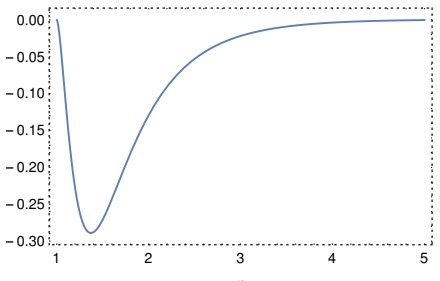
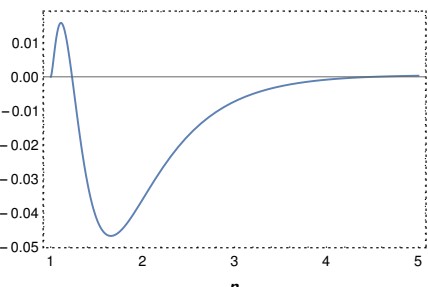

**Figure 2.** Behavior of the "effective" strong energy condition (**left**) and "effective" null energy (**right**) condition for $\omega = -1.43$ represented in the left-hand side of Equations (33) and (34), taking into account the effects of the non-minimal coupling.

It is important to observe that only in the case of radiative fluid is it possible to obtain a model without singularity preserving the energy conditions, at least in the BD theory. It

is true also for the model with flat spatial sections. For a non-flat universe, one can obtain a singularity-free scenario even in General Relativity if the strong energy condition—but not necessarily the null energy condition—is violated.

## 4. Canonical Quantization of the BD Theory

In the early Universe, when the cosmo is compressed in a Planck scale, quantum effects become relevant, and the quantization of the classical models may be a necessity. Quantum cosmology stands on this principle, considering a wave description of the universe that satisfies the Hamiltonian constraint of the theory. In this section, we will investigate the canonical quantization of the BD theory in the minisuperspace, considering the FLRW metric (18). We shall quantize the Hamiltonian constraint $\mathcal{H}_{\text{tot}} \approx 0$ (where "$\approx$" means weak equality. See, for example, [36]) to obtain the Wheeler–DeWitt Equation $\hat{H}_{\text{tot}}\Psi = 0$, with $\Psi$ being the wave function of the universe, and using the canonical operators ($\hbar = 1$),

$$x \rightarrow \hat{x} : \psi(x) \rightarrow x\psi(x) \quad ; \quad \pi_x \rightarrow \hat{\pi}_x : \psi(x) \rightarrow -i\partial_x \psi(x). \tag{35}$$

The total Hamiltonian $\mathcal{H} = \mathcal{H}_{\text{tot}}$ is formed by the gravitational part and the Hamiltonian of the matter content, $\mathcal{H} = \mathcal{H}_G + \mathcal{H}_M$. To find $\mathcal{H}_M$, let us consider a model of early-universe filled with a radiative fluid. Using Schutz formalism [6], the super-Hamiltonian of the fluid is given by

$$\mathcal{H}_M = \frac{N}{a}\pi_T, \tag{36}$$

where $T$ is directly related to the entropy of the fluid [37]. With this, the canonical quantization of the total Hamiltonian in the original Jordan's frame is

$$\partial_a^2 \Psi + \frac{p}{a}\partial_a \Psi + \frac{6}{\omega}\frac{\phi^2}{a^2}\left\{\frac{a}{\phi}\partial_a\partial_\phi \Psi - \left(\partial_\phi^2 \Psi + \frac{q}{\phi}\partial_\phi \Psi\right)\right\} = -12i\frac{(3+2\omega)}{\omega}\phi\partial_T \Psi, \tag{37}$$

where $p, q$ are ordering factors for the quantization of the momenta squared. For more details on the computation of (37), see Ref. [37]. Notice, however, that here we use a different sign convention.

The calculation is similar in Einstein's frame, considering the transformation (8). In this case, the canonical quantization results in

$$\partial_b^2 \Psi + \frac{\bar{p}}{b}\partial_b \Psi - \tilde{\omega}\frac{\phi^2}{b^2}\left\{\partial_\phi^2 \Psi + \frac{\bar{q}}{\phi}\partial_\phi \Psi\right\} = -i\partial_T \Psi. \tag{38}$$

Here, again, we have that $\bar{p}$ and $\bar{q}$ are ordering factors.

Notice that Equations (37) and (38) are Schrödinger-like, that is,

$$\hat{H}\Psi = i\frac{\partial}{\partial t}\Psi, \tag{39}$$

if we consider the matter field playing the role of time. We can, therefore, treat it as a quantum system to analyze this cosmological scenario in the early-universe. The first step is to verify the conditions for the effective Hamiltonian operators of these models to be self-adjoint. It is known [37] that the Hamiltonian operator of this quantized BD model with a radiative fluid can be self-adjoint only if $q = \bar{q} = 1$. On the other hand, we have the quantum equivalence between Jordan's and Einstein's frames [38] if, and only if, $p = \bar{p} = 1$, and thus we can use Einstein's frame.

Now, in Einstein's frame, we choose the coordinate

$$\phi = e^{\sqrt{\frac{12}{|3+2\omega|}}\sigma} \tag{40}$$

instead of $\phi$. With this, the relation between the scale factor in Jordan's and Einstein's frames is

$$a = e^{-\frac{\sigma}{\sqrt{|1+\frac{2}{3}\omega|}}} b. \tag{41}$$

In terms of the new variable $\sigma$, the Schrödinger Equation (39) is

$$\partial_b^2 \Psi + \frac{1}{b}\partial_b \Psi - \epsilon \frac{1}{b^2}\partial_\sigma^2 \Psi = -i\partial_T \Psi. \tag{42}$$

The measure of the Hilbert space is such that

$$\langle \psi | \psi \rangle = \int_{-\infty}^{\infty} \int_0^{\infty} \psi\psi^* b\, db d\sigma. \tag{43}$$

The regular solution for the Equation (42) is

$$\Psi(b,\sigma) = A(k,E)J_\nu(\sqrt{E}b)e^{i(k\sigma - ET)}, \quad \nu = \sqrt{-\epsilon}|k|, \tag{44}$$

where $k$ is a separation constant. There is also another solution written in terms of the Bessel function $J_{-\nu}(x)$, which is not regular at the origin, at least when $\epsilon = -1$. For this reason, we will disregard it at the moment. We will choose $\epsilon = -1$; for this case, the Hamiltonian operator is bounded from below, and it is essentially self-adjoint for $-1 \le \bar{p} \le 3$ (see [37]), which is our case. Thus, $\nu = |k|$. Let us choose the coefficient $A(k,E)$ such that the wave packet becomes

$$\Psi(b,\sigma,T) = \frac{1}{\mathbf{N}} \int_0^{\infty} \int_{-\infty}^{+\infty} e^{-k^2} x^{|k|+1} e^{-\alpha x^2} J_{|k|}(xb)\, e^{ik\sigma} dkdx, \tag{45}$$

where $\mathbf{N}$ is a normalization factor, and

$$\alpha = \gamma + iT, \quad x = \sqrt{E}, \tag{46}$$

with $\gamma$ a positive real parameter. The integration in $x$ gives us [39],

$$\Psi(b,\sigma,T) = \frac{1}{\mathbf{N}} \int_{-\infty}^{+\infty} e^{-k^2 + ik\sigma} \frac{b^{|k|}}{(2\alpha)^{|k|+1}} e^{-\frac{b^2}{4\alpha}} dk. \tag{47}$$

To find the normalization factor $\mathbf{N}$, we should remember that

$$\langle \Psi, \Psi \rangle = \int_0^{\infty} \int_{-\infty}^{\infty} \Psi\Psi^* b\, db d\sigma = 1, \tag{48}$$

and thus

$$\mathbf{N}^2 = \int_0^{\infty} \int_{-\infty}^{\infty} \left[ \int_{-\infty}^{\infty} e^{-k^2} \frac{b^{|k|}}{(2\alpha)^{|k|+1}} e^{-\frac{b^2}{4\alpha}} e^{ik\sigma} dk \right]$$
$$\cdot \left[ \int_{-\infty}^{+\infty} e^{-k'^2} \frac{b^{|k'|}}{(2\alpha^*)^{|k'|+1}} e^{-\frac{b^2}{4\alpha^*}} e^{-ik'\sigma} dk' \right] bdb\, d\sigma.$$

Integrating over $\sigma$, $k'$, and $k$, and defining $u = b/2|\alpha|$, we obtain

$$\mathbf{N}^2 = 4\pi\sqrt{\frac{\pi}{8}} \int_0^{\infty} u^2 e^{-\gamma u^2} \left[1 - \Phi\left(\frac{\ln u}{\sqrt{2}}\right)\right] du$$
$$= 4\pi\sqrt{\frac{\pi}{8}} \left[\frac{\sqrt{\pi}}{4\gamma^{\frac{3}{2}}} - g_{(1)}(\gamma)\right],$$

where $\Phi(x)$ is the error function, and

$$g_{(1)}(\gamma) = \int_0^\infty u^2 e^{-\gamma u^2} \Phi\left(\frac{\ln u}{\sqrt{2}}\right) du. \tag{49}$$

Since $-1 < \Phi(x) < 1$ and $\int_0^\infty u^2 e^{-\gamma u^2}$ is strictly positive, we have

$$|g_{(1)}(\gamma)| < \frac{\sqrt{\pi}}{4\gamma^{\frac{3}{2}}}. \tag{50}$$

Therefore, $\mathbf{N}^2$ is positive, as expected.

We can now calculate the expected values of the scale factor $b$ and scalar field $\sigma$. For the scale factor,

$$\langle b \rangle = \langle \Psi | b | \Psi \rangle. \tag{51}$$

Similarly to the calculation of the normalization factor, integrating in $\sigma$ and $k'$, we have

$$
\begin{aligned}
\langle b \rangle &= \frac{8\pi|\alpha|}{\mathbf{N}^2} \sqrt{\frac{\pi}{8}} \int_0^\infty u^3 e^{-\gamma u^2} \left[1 - \Phi\left(\frac{\ln u}{\sqrt{2}}\right)\right] du \\
&= \frac{8\pi|\alpha|}{\mathbf{N}^2} \sqrt{\frac{\pi}{8}} \left[\frac{1}{2\gamma^2} - g_{(2)}(\gamma)\right],
\end{aligned}
$$

where

$$|g_{(2)}(\gamma)| = \left| \int_0^\infty u^3 e^{-\gamma u^2} \Phi\left(\frac{\ln u}{\sqrt{2}}\right) du \right| < \frac{1}{2\gamma^2}. \tag{52}$$

Since $|\alpha| = \sqrt{\gamma^2 + T^2}$, and defining

$$\Omega(\gamma) = \frac{8\pi}{\mathbf{N}^2} \sqrt{\frac{\pi}{8}} \left[\frac{1}{2\gamma^2} - g_{(2)}(\gamma)\right] = 2 \left[\frac{\gamma^{-2} - 2g_{(2)}(\gamma)}{\frac{\sqrt{\pi}}{2\gamma^{\frac{3}{2}}} - 2g_{(1)}(\gamma)}\right], \tag{53}$$

which is strictly positive, the expected value of $b$ is

$$\langle b \rangle = \Omega(\gamma) \sqrt{\gamma^2 + T^2}. \tag{54}$$

By definition, $\gamma > 0$. Therefore, $\langle b \rangle > 0$, that is, there is no singularity at $T = 0$. Moreover, for $T \gg \gamma$, at late times, $\langle b \rangle \to T$.

Using the fact that $\int f(x)\delta'(x - x_0)dx = -\int f'(x)\delta(x - x_0)dx$, a similar calculation for $\sigma$ results in

$$\langle \sigma \rangle = -\frac{2\pi i}{\mathbf{N}^2} \int_0^\infty \int_{-\infty}^{+\infty} e^{-k^2} \frac{b^{|k|+1}}{(2\alpha)^{|k|+1}(2\alpha^*)} e^{-\left(\frac{\gamma b^2}{4|\alpha|^2}\right)} \partial_k \left[e^{-k^2} \frac{b^{|k|}}{(2\alpha^*)^{|k|}}\right] dk db.$$

The integral over $k$ is zero because the function is odd with respect to this parameter. Thus,

$$\langle \sigma \rangle = 0. \tag{55}$$

This does not mean, however, that the scalar field $\phi$ is a constant, since the expectation value allows fluctuations. Notice that we have a symmetrical bouncing in both frames because of Equation (41).

## 5. Analysis of the Solution via the de Broglie–Bohm Approach

In the previous section, we introduced a quantum model of the BD theory, identifying the universe with a wave function that obeys a Schrödinger-like equation. Thus, we proceeded with the usual methods of quantum mechanics to analyze the behavior of the scale factor, which is directly connected with the volume of the universe. However, in the

case of quantum cosmology, we must give up on the usual Copenhagen interpretation of QM, since it requires an external observer to cause the wave function to collapse into one state. An alternative is the many-world interpretation, which considers every state of the wave function as real, existing in parallel with each other. Our universe, therefore, is one of many. This interpretation does not require the collapse of the wave function, and so we can investigate different states separately from the wave packet.

The de Broglie–Bohm (dBB) interpretation, on the other hand, does not consider a wave description of the universe at all. Instead, it is a dynamical theory in which real trajectories can be obtained in the configuration space of the quantum system. Those trajectories are observer-independent, and, therefore, the de Broglie–Bohm interpretation does not rely on any collapse mechanism. This interpretation was formulated as a causal alternative to the probabilistic Copenhagen interpretation still in the early years of the quantum era [40,41], and it has replicated most of the classic results of quantum mechanics. In the dBB interpretation, observers are also described by quantum operators to be applied on a wave function that satisfies the Schrödinger equation. The wave function dictates the equations of motion to find the trajectories. In Bohmian mechanics, $\Psi(x_i, t)$ is decomposed as

$$\Psi(x_i, t) = R(x_i, t) e^{iS(x_i, t)} \tag{56}$$

and the probability density of the trajectory is given by $\rho(x_i, t) = |\Psi(x_i, t)| = R^2(x_i, t)$. The Bohmian trajectories are realized through the momenta defined as

$$p_j = \partial_j S = \left( \frac{i}{2} \right) \frac{\Psi \Psi^*_{,j} - \Psi^* \Psi_{,j}}{|\Psi|^2}. \tag{57}$$

Since it is not probabilistic, the observers in the dBB do not necessarily need to be represented by self-adjoint operators.

This ontological interpretation of the quantum theory has its critics, who argue that other interpretations may be better-suited [42], or take issue with the so-called hidden variables which determine the behavior of the trajectories in a many-body configuration [43]. Still, because of its distinct characteristic, the dBB interpretation is a suitable candidate to be applied in the quantization of cosmological scenarios [44,45], where the wave function becomes a guide to the possible observer-independent evolution of the universe. Notice that, contrary to the Copenhagen interpretation, the dBB interpretation of quantum mechanics does not necessarily need to work in the Hilbert space.

Let us compare these two approaches in particular cases. In the previous section, we have already made the computation using the expectation values finding, quite generally, that the expectation value for the scalar field is zero and the expectation value for the scale factor is the same as found in the corresponding case in GR. We will now analyze the predictions for the evolution of the universe using the dBB formulation. We will work initially in the Einstein frame. Remember the usual solution to the Equation (42) given in Equation (44):

$$\Psi(b, \sigma, T) = A(k, E) J_\nu(\sqrt{E}b) e^{i(k\sigma + ET)}, \quad \nu = \sqrt{-\epsilon}|k|. \tag{58}$$

The construction of the wave packet exposed in the previous section is not convenient here. The reason is that the integration is made in the interval $-\infty < k < \infty$, but the terms coming from the order of the Bessel functions depend on $|k|$. Hence, the integration on the whole interval does not allow for obtaining a tractable form for the wave packet in view of using the de Broglie–Bohm approach. Let us consider the general wave packet given by

$$\Psi(b, \sigma, T) = \int_0^\infty \int_{-\infty}^{+\infty} A(k) x^{\nu+1} e^{-(\gamma + iT)x^2} J_\nu(xb) e^{ik\sigma} dk dx, \tag{59}$$

with the definition $x = \sqrt{E}$ and $\nu = |k|$. The integration in $x$ leads to

$$\Psi(b,\sigma,T) = \int_{-\infty}^{+\infty} A(k) \frac{b^\nu}{(\alpha)^{\nu+1}} e^{-\frac{b^2}{4\alpha}} e^{ik\sigma} dk. \tag{60}$$

With this, we investigate the behaviors of the scale factor and the scalar field in some special cases, and compare them with the results of the previous section.

### 5.1. The Scalar Field Is Absent

Let us consider the case where the factor $A$ is given by a delta function. If we choose $A(k) = \delta(k)$, which means the world where $\nu = 0$, the contribution of the scalar field vanishes and the wave function is such that

$$\Psi(b,\sigma,T) = \frac{e^{-\frac{b^2}{4\alpha}}}{\alpha}. \tag{61}$$

Notice that this does not mean we recover General Relativity. This would be one world of the many in a BD theory of gravitation. Considering Equation (48), it is straightforward to verify that the norm of the wave function (61) is finite and time-independent. Similarly to the calculation in the last section, we can compute the expected value of the scale factor:

$$< b >= \frac{1}{\mathbf{N}^2} \int_0^\infty \frac{e^{-\gamma \frac{b^2}{2|\alpha|^2}}}{|\alpha|^2} b\, db = \frac{1}{\gamma \mathbf{N}^2} \sqrt{\gamma^2 + T^2}, \tag{62}$$

where $\mathbf{N}$ here is the normalization factor of the wave function (61), naturally. Therefore, in this world, a bounce occurs, and, for late times, that is, when $T \gg \gamma$, $\langle b \rangle \to T$.

For the Bohmian trajectories (57), we use the phase of the wavefunction (61), which is, in this case,

$$S = T \frac{b^2}{4|\alpha|^2} - \arctan\left(\frac{T}{\gamma}\right). \tag{63}$$

Remembering that the conjugate momentum is $p_b = \dot{b}/2$, we obtain for the Bohmian trajectories,

$$\dot{b} = \frac{bT}{|\alpha|^2}. \tag{64}$$

The solution of this differential equation is $b = b_0 \sqrt{\gamma^2 + T^2}$, which is the same result as before using expectation values. In a Bohmian analysis, this universe also has a bounce.

The results exposed in the previous section show quite generically that wave packets with constant finite norm lead to a zero expectation value for $\sigma$. From now on, we explore possibilities where we circumvent this restriction but at the price of having wave packets that are not finite, what is not an obstacle when dBB formulation is used, see Ref. [11].

### 5.2. A Single Scalar Mode

Let us consider now the superposition function such that $A(k) = \delta(k - k_0)$, where $k_0$ is a positive constant. This implies to consider a single scalar mode behaving like a plane wave. The wave function reads as

$$\Psi(b,\sigma,T) = \frac{b^{\nu_0}}{(\alpha)^{\nu_0+1}} e^{-\frac{b^2}{4\alpha}} e^{ik_0\sigma}, \tag{65}$$

with $\nu_0 = \sqrt{-\epsilon}k_0$. We will analyze the Bohmian scenario for both $\epsilon = 1$ and $\epsilon = -1$.

Let us start with $\epsilon = -1$. In this case, the phase of the wave function becomes

$$S = \frac{b^2}{4|\alpha|^2} T - (k+1)\arctan\left(\frac{T}{\gamma}\right) + k_0\sigma. \tag{66}$$

The presence of the term $k_0\sigma$ changes the previous analysis, since now the variable $\sigma$ is featured in the wave function. The guidance equations become:

$$\dot{b} = \frac{bT}{|\alpha|^2}, \tag{67}$$

$$\dot{\sigma}b^2 = 12k_0, \tag{68}$$

and their solutions are,

$$b = b_0\sqrt{\gamma^2 + T^2}, \tag{69}$$

$$\sigma = \sigma_0 \arctan\left(\frac{T}{\gamma}\right). \tag{70}$$

The scalar field is no longer a constant. In the original Jordan frame, Equation (69) yields

$$a \propto \exp\left[-\frac{\arctan\left(\frac{T}{\gamma}\right)}{\sqrt{|1 + \frac{2}{3}\omega|}}\right]\sqrt{\gamma^2 + T^2}. \tag{71}$$

Again, we have a non-singular solution recovering the classical solution asymptotically, but now the bounce in this case is asymmetric. Asymmetric bouncing models have been studied in Ref. [46].

Now, for the case $\epsilon = +1$, in an observer-dependent interpretation, this becomes a problematic situation since the energy $E$ is not bounded from below. The signature of the Schrödinger equation is hyperbolic instead of elliptic. However, we can follow the same step as before to obtain the phase of the wave function and to compute the Bohmian trajectories. The main new feature now is that the order of the Bessel function is imaginary, $\nu = ik$. Hence, the phase of the wave function (60) for this case is

$$S = k_0 \ln\frac{b}{|\alpha|^2} + \frac{b^2}{4|\alpha|^2}T - \arctan\left(\frac{T}{\gamma}\right) + k_0\sigma. \tag{72}$$

The equations for the Bohmian trajectories become

$$\dot{b} = \frac{2k_0|\alpha|^2}{b} + \frac{T}{|\alpha|^2}b, \tag{73}$$

$$\dot{\sigma} = -12\frac{k_0}{b^2}. \tag{74}$$

The solutions are

$$b = b_0\sqrt{\gamma^2 + T^2}\sqrt{\sigma_0 - \bar{k}_0 \arctan\left(\frac{T}{\gamma}\right)}, \tag{75}$$

$$\sigma = 3\ln\left\{\sigma_0 - \bar{k}_0 \arctan\left(\frac{T}{\gamma}\right)\right\} \tag{76}$$

where $b_0$ and $\sigma_0$ are constants and $\bar{k}_0 = 36k_0/(b_0^2\gamma)$.

To return back to the scale factor in the Jordan frame, we use

$$a = \phi^{-1/2}b = e^{-\frac{\sigma}{\sqrt{1 + \frac{2}{3}\omega}}}. \tag{77}$$

Hence,

$$a = a_0\sqrt{\gamma^2 + T^2}\left\{\sigma_0 + \bar{k}_0 \arctan\left(\frac{T}{\gamma}\right)\right\}^r, \tag{78}$$

with

$$r = \frac{1}{2} \frac{\sqrt{1 + \frac{2}{3}\omega + 6}}{\sqrt{1 + \frac{2}{3}\omega}}. \tag{79}$$

Notice that the solutions (75) and (76) are non-singular only if $\sigma_0 > \frac{\pi}{2}\bar{k}_0$ and $\sigma_0 > 0$. The classical solution is recovered asymptotically. This fact reveals again the very peculiar features when the energy conditions are satisfied.

We can compute the quantum potential that results from the modified Hamilton–Jacobi equation in the dBB formulation of quantum mechanics [11], which is given by

$$V_Q = \frac{\nabla^2 R}{R}, \quad R = \sqrt{\Psi^* \Psi}. \tag{80}$$

The Laplacian operator is defined in the minisuperspace with variables $b$ and $\sigma$. For the cases studied in this subsection, the result is the expected: the quantum potential reaches its maximum value at the bounce, and decreases to zero asymptotically where the classical solutions are recovered.

In performing the study above, we have used the Bessel function given in a (44). It can be explicitly verified that, if we have used the Bessel function of negative order, which is not regular at the origin, the results would be the same with a reversal of the time, $T \to -T$.

### 5.3. Multiple Scalar Modes

We can combine different scalar modes. One example can be achieved by the combination,

$$A(k) = \delta(k - k_0) + \eta \delta(k + k_0), \tag{81}$$

with $\eta = \pm 1$. With a similar computation, we obtain, for the positive sign, $\cos k_0 \sigma$ instead of $e^{ik_0}$ and for a negative sign $\sin k_0 \sigma$. In both of these cases, the scalar field is not present in the phase of the wave function, and we recover the same solutions already given in the case that the scalar field is absent.

## 6. Conclusions

The Brans–Dicke theory is one of the oldest proposals of a modification to the theory of general relativity. Although it has been studied for over sixty years, the Brans–Dicke theory continues to reveal intriguing aspects. Some of these were discussed in the present work concerning classical and quantum scenarios for the early universe. First of all, it is possible to obtain a singularity-free cosmological solution if the Brans–Dicke parameter $\omega$ varies as $-3/2 < \omega < -4/3$. In this range, the energy conditions are satisfied in the Einstein frame: the avoidance of the singularity is driven by the non-minimal coupling.

When we turn to the quantum scenario in the minisuperspace, other curious features appear. The energy is bounded only for $\omega < -3/2$, that is, when the energy condition is violated in the Einstein frame. On the other hand, in the case of $\omega > -3/2$, the energy conditions are satisfied, but the energy is not bounded from below, so it becomes problematic to employ the usual interpretation scheme based on the Copenhagen formulation of quantum mechanics: the wave function is not finite anymore.

This issue motivated us to consider the de Broglie–Bohm interpretation of quantum mechanics. The previous results are found again if $\omega < -3/2$. However, when $\omega > -3/2$, we have either a singular or non-singular solution. This implies that, in the interval $-\frac{3}{2} < \omega < -\frac{4}{3}$, the classical model displays singularity-free scenarios, while the quantum models may display either singular or non-singular solutions. This result raises the following question: can such results occur in more general modified gravity theories belonging to the Horndesky class?

We add to the previous discussion some additional remarks. First of all, it must be verified to what extent the results reported here depend on the wave packet construction and the choice of the time variable. On the other hand, the problems of convergence of the wave function, when the energy conditions are obeyed, is somehow general in the presence of scalar fields since it leads to a hyperbolic signature in the Hamiltonian and, consequently, in the Schrödinger-type equation. However, such a well-known fact acquires new features that were briefly described above. The analysis displayed in the present work shows that the well-studied Brans–Dicke cosmological models present interesting properties at classical and quantum levels as well, no matter which interpretation is chosen.

**Author Contributions:** All authors contributed equally to the present work. All authors have read and agreed to the published version of the manuscript.

**Funding:** This research was supported by the National Scientific and Technological Research Council (CNPq, Brazil) and the State Scientific and Innovation Funding Agency of Espírito Santo (FAPES, Brazil).

**Conflicts of Interest:** The authors declare no conflict of interest.

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
