# Peer review of "Quantum and Classical Cosmology in the Brans–Dicke Theory"

_universe, doi:10.3390/universe7080286_

Round 1

Reviewer 1 Report

In my opinion, the review paper is solid but can be improved in the sense of enlarging it a bit. On the one hand, there are papers by PNeto and Colistete and Moniz-Marto about Broglie -Bohm quantum cosmology and those could be mentioned and also explicitly appraised. Likewise from P Peter, and many others. On the other hand, this may be just the sole review on this SI that focuses on Broglie-Bohm quantum cosmology, hence the authors could indeed include more features. I do not mean so to repeat what other papers and surely reviews convey but possibly add more context, plus content. Here is an opportunity. 

A few more technical and possible unaddressed questions but that could be interesting to have the authors say or at least speculate (soberly) about it.: Can BD (Brans Dicke)  (due to its more Machian context) overcome any of the difficulties that Broglie-Bohm (BB) has in plain GR and the singularities of the quantum potential? Are quantum potentials improved i.e. easier to employ in scattering or bouncing if the parameter is free to tune? In other words, how does BD in the BB scheme improve or not,  contrasted to plain GR? I am also thinking of the scattering analogy of the pre-big-bang framework develop by Veneziano and Gasperini (VG) and how BB changed a few things. I am aware of section 3 regarding bouncing solutions. I was just thinking if a sort of 'universe creation' as per VG construction could be possible. Or not, and why. Now, take BD and some of that scattering lines. Can any quick ideas be brought to speculate and explore, even if marginally now? 

I would thus point that the paper can be accepted but it would be a far much better contribution it either addressing or trying to address the points above, constructively. 

Reviewer 2 Report

The authors review and present new results concerning the classical and quantum treatment of cosmological models in Brans-Dicke theory.

The paper is nicely presented and motivated. The quantum scenario, using a canonical quantization procedure, is discussed and indeed lead to novel interesting aspects concerning these cosmological scenarios. I believe that the results presented in this paper, complemented with a set of interesting questions still to be addressed should be interesting to the reader. Therefore, I recommend publication of this article.

Reviewer 3 Report

This is a paper which is supposed to be a review paper and not a research article. In view of that I have lots of criticism which follows:

1. The literature to the problem is very poor as for a ''review''. A lot of important reference related to classical and quantum Brans-Dicke (BD) theory are missing (e.g. seminal paper by Pascual Jordan of 1959 which was published before Brans and Dicke paper).

2. In relation to that, if the Authors wanted to keep it as it is, I think that the paper would have the title be changed by adding sth like ''..in canonical and Bohmian approach'' though this still does not kill my above comment about the poorness of the literature.

3. Not mentioning much much richer and appropriate literature of the subject, some important pieces of Brans-Dicke related/inspired topics are totally omitted. For example, there is no mention of a very important in the literature studies the specific case of string cosmology, which is BD for the parameter ω = -1. In fact, these were achievements of this cosmology which are explored in the article – what I mean are the dual solutions/branches.

4. Besides, there is no mention or deeper discussion of the case of ω = -3/2 which is the conformally invariant cosmology deeply studied in the literature (following P. Jordan). Its peculiarity can be seen for example in the transformation formulas (33) and (34) which in the limit leave σ and b arbitrary while Φ and a go to infinity and zero respectively (by the way also leaving r.h.s of (30) vanishing which does not seem to be a problem). There is also a problem with Eq. (64) in this context.

5. In general, I do not see any definition of the quantum avoidance of the classical singularity (e.g. vanishing of the wave function or its module – see e.g. Ref. [3]). Singularity avoidance or a ''bounce'' is mentioned but in my opinion (a “review”), not properly defined. The readers should have some benefit after reading the paper.

6. There are some small issues related to English and terms used. The term before Eq. (44) - '' the SCALAR factor'' sounds ugly to a cosmologist or even more general audience – standard name is ''the SCALE factor'' unless something else is meant. Also, more appropriate is ''a bounce” rather than ''a bouncing''; ''a reversal” rather than “a reversion'', ''Bohmian'' instead of “BoHemian'' etc. In Eq. (30) two derivatives do not act on the wave function (a typo). I do not see the reason why Eqs. (37) and (51) are repeated?

I think the paper requires a very deep revision before it can be considered for publication.

Reviewer 4 Report

The manuscript entitled “Quantum and classical cosmology in the Brans-Dicke theory ” is devoted to exploring the classical and quantum aspects of cosmological models in Brans-Dicke theory. In this manuscript, the author first reviews cosmological bounce solutions in Brans-Dicke theory and quantizes this classical model in a canonical way. Then they studied both the classical and quantum solutions of the Brans-Dicke theory in a spatially flat FLRW universe filled with radiation. The paper is well organized and I believe the study presented in this manuscript warrants publication. Before this paper can be published, I would like to present here some minor comments that could be considered by the authors to improve the manuscript.

First, there are a few typos in the draft.

(1) In line 79, “non-minimal coupled ” → “non-minimally coupled”.

(2)Part of the right panel in Figs. 1-2 are not visible.

(3) line 155 “Using the Friedmann equations” → “Using the Friedmann and the Raychaudhuri equations” • Line 107 “violated” → “violate”

(4)Below lIne 208, there should be a minus sign in the formula behind “using the fact that”

(5)Line 248 “A bouncing occurs” → “A bounce occurs”

Besides, I also have some questions

(6)For solutions (21)-(22), is there any regime during evolution with a Planck-scale curvature, or what is the maximum energy density for these solutions?

(7)In Line 123 the authors said “at the early stages of the expansion of the universe the curvature s not essential”, does that mean the bounce happens at a low curvature regime?

(8)In general, we believe the quantum theory can remove the singularities inherent in the classical model. However, for −3/2 < w < −4/3, it turns out that the classical solutions are non-singular while the corresponding quantized model can have singular solutions. Can the authors add more comments in the conclusion on what is the possible physical implications of the singularity in the quantum solutions?

Round 2

Reviewer 3 Report

The Authors took seriously my criticism and improved the paper along the lines I suggested. It can be published now.